# Metal Toxicity: Effects on Energy Metabolism in Fish

**DOI:** 10.3390/ijms25095015

**Published:** 2024-05-03

**Authors:** Natalia A. Gashkina

**Affiliations:** Vernadsky Institute of Geochemistry and Analytical Chemistry, Russian Academy of Sciences, 19 Kosygin St., Moscow 119991, Russia; ngashkina@gmail.com

**Keywords:** metal toxicity, energy metabolism, mitochondria, ATP and ROS production, hypoxia, anaerobic metabolism

## Abstract

Metals are dispersed in natural environments, particularly in the aquatic environment, and accumulate, causing adverse effects on aquatic life. Moreover, chronic polymetallic water pollution is a common problem, and the biological effects of exposure to complex mixtures of metals are the most difficult to interpret. In this review, metal toxicity is examined with a focus on its impact on energy metabolism. Mechanisms regulating adenosine triphosphate (ATP) production and reactive oxygen species (ROS) emission are considered in their dual roles in the development of cytotoxicity and cytoprotection, and mitochondria may become target organelles of metal toxicity when the transmembrane potential is reduced below its phosphorylation level. One of the main consequences of metal toxicity is additional energy costs, and the metabolic load can lead to the disruption of oxidative metabolism and enhanced anaerobiosis.

## 1. Introduction

Metals, as ubiquitous elements in the environment, are dispersed and accumulate in natural environments (particularly aquatic environment) due to anthropogenic activities, causing environmental and public health problems in many places around the world [1,2,3,4,5]. Heavy metals are elements that have an atomic mass greater than 20. Nonessential metals, such as mercury (Hg), thallium (Tl), cadmium (Cd), tin (Sn), and lead (Pb), have no proven biological functions. The essentiality of macrominerals (calcium (Ca), phosphorus (P), magnesium (Mg), sodium (Na), potassium (K), and chloride (Cl)) and certain elements (cobalt (Co), copper (Cu), iodine (I), iron (Fe), manganese (Mn), selenium (Se), and zinc (Zn)) has been confirmed in fish [6]. Essential metals are required in trace amounts for biological life due to their participation in metabolic reactions as cofactors or integral parts of enzymes, but they become toxic at high levels, and non-essential metals can be toxic even in trace amounts, disturbing biological processes [7]. While the effects of individual metals are well understood, identifying the effects of complex mixtures of metals is a more difficult task.

The long-standing hypothesis that bioenergetics (a field in biochemistry and cell biology that concerns energy flow through living systems), respiratory physiology, and fish metabolism should be of particular importance for their ecological performance is gaining relevance and can be used in the context of a new emerging field of conservation physiology [8]. Moreover, bioenergetics, viz., the redistribution of energy depending on the needs of the organism, can be considered with important potential trade-offs, particularly those dealing with survival [9].

Metabolism is a highly coordinated cellular activity in which many multienzyme systems (metabolic pathways) interact to produce chemical energy, the cell’s own characteristic molecules, and the synthesis and degradation of biomolecules necessary for specialized cellular functions. The application of adverse outcome pathways (i.e., the linking of pollutant exposure via early molecular and biochemical changes to physiological effects) has indicated that several key events requiring energy for stress responses and toxic defense are likely to converge in a single common event, resulting in increased metabolic demands [10]. Biological effects from exposure to complex mixtures may go undetected; therefore, the strength of metabolomics is its ability to identify mixtures affecting the physiology of fish without prior assumptions about the nature and mode of action of the chemicals within the mixtures [11]. In addition, the redistribution of ions in organs and tissues and the bioaccumulation of metals, especially essential ones, may indicate alterations in metabolic fluxes [12].

The purpose of this review was to consider the impact of metal toxicity on energy metabolism and mechanisms regulating adenosine triphosphate (ATP) production and reactive oxygen species (ROS) emission, and its role in the development of catabolic situations, hypoxia, and anaerobic metabolism.

## 2. Metals in the Environment

Metals (macro- and microelements) are natural components of the environment (lithosphere, hydrosphere, atmosphere, and biosphere). However, human activities have further caused a gradual increase of metals in the atmosphere, water, and soil [1]. Emerging contaminants can be increasingly associated with the waste and wastewater resulting from industrial, agricultural, or municipal activities that create environmental and public health problems in many locations around the world [3,4,5]. In addition to toxic water pollution with metals, human activities can lead to the acidification and eutrophication of the aquatic environment, which increases the bioavailability of metals and worsens the oxygen regime of the habitat environment [13].

Although the implementation of environmental regulations has reduced metal pollution in many water bodies, aquatic organisms are still chronically exposed to elevated metal concentrations. Examples of viable populations that have been heavily exposed to metal contamination over the past century include the brown trout (*Salmo trutta*) living in the River Hayle in Cornwall (southwest England) [14], wild yellow perch (*Perca flavescens*) from five lakes in an area of industrial Cu extraction in Sudbury, Ontario [15], *Hyphessobrycon luetkenii* from historically contaminated sites of the João Dias creek in Minas do Camaquã, Brazil [16], and whitefish (*Coregonus lavaretus* L.) from historically contaminated subarctic Lake Imandra [17,18].

Even abandoned mining sites can cause metals to accumulate in natural environments, causing adverse effects on aquatic life and human health [5,19]. For example, modeling of the spread of metal pollution shows that even after the shutdown of a non-ferrous-metals smelting plant, pollution can affect the aquatic environment for many decades due to the accumulated metals in soils and bottom sediments [20].

Although “pollution hotspots” remain, the chronic polymetallic pollution of natural environments is more widespread around the world. For example, even treated wastewater contains complex mixtures of micropollutants. Thus, metal pollution of the environment, particularly the aquatic environment, will remain a vital problem in this century.

## 3. Metal Toxicity

A review of the chemical–biological interactions of essential and toxic elements summarizes the processes that cause metal toxicity [21]:(1)They interact competitively with the elements which are necessary for proper functioning of the body (competition for binding sites, e.g., on transporter proteins, enzymes, nucleic acids, neurotransmitters, sugars, ATP, glutathione);(2)They block the functional groups of significant biologically active compounds, e.g., -SH, -OH, -NH2, -COOH, and -S-S;(3)They change the structure of biomolecules and biological membranes;(4)They limit the pool of available bioanions (e.g., phosphates);(5)They interfere with the synthesis of ATP;(6)They form insoluble salts in biological fluids;(7)They participate in redox reactions that generate free radicals.

Metal toxicity depends on the level and duration of exposure, which largely determine the consequences of exposure to metals.

Metal accumulation in the body affects enzymatic and metabolic activity and leads to the following consequences [22,23,24]:(1)Oxidative stress and lipid peroxidation (reactive oxygen species (ROS) increase, membrane damage, and oxidation of biomolecules);(2)DNA damage (mutagenicity, cell death, impaired DNA replication);(3)Enzyme disfunction and metabolic disorders (cell mechanism disturbances, metabolic disruptions, organ and tissue pathology);(4)Endocrine disruption (hormonal production disturbances, alteration in reproductive parameters).

Due to their properties, metals can also have specific effects, for example Hg has neurotoxicity, Cd and Ni have nephrotoxicity, Pb has hemotoxicity, As significantly affects hematology and immunology, and Cu affects osmoregulation [23,24,25,26].

## 4. Energetic and Metabolic Effects of Metal Toxicity

The metabolic theory of ecology takes as its basis the concept of basal or standard metabolic rate (the amount of energy that fish require to simply stay alive), which is a convenient starting point for estimating the energy costs of physiological adaptation to the environment [27,28]. Furthermore, the basal metabolic rate does not take into account the costs of growth and reproduction, whereas locomotion can increase it by up to three- to five-fold in most fish species [29].

Habitat toxicity can also increase energetic costs: using classical techniques of swimming respirometry, the energy costs of fish organismal maintenance were found to be higher at the most contaminated sites in field experiments [10].

Thus, body oxygen consumption, energy flows, and their redistribution are affected by habitat (temperature, food supply, etc.) and fish physiological conditions (maturating stage, starvation, etc.) including locomotion, as well as the metal toxicity of the environment (which, via oxidative stress, detoxification and other processes, increases additional energy costs) (Figure 1).

### 4.1. Oxygen Consumption and ATP Production

It is estimated that 90% of cellular oxygen consumption occurs in the mitochondria (~20% is consumed by mitochondria to counteract the mitochondrial proton leak, and the remaining 70% is consumed for mitochondrial adenosine triphosphate (ATP) production) [28,30].

Mitochondria are well-known organelles that rely on oxidative phosphorylation to produce ATP; ATP is synthesized from glucose by processing through glycolysis and the tricarboxylic acid cycle (TCA) to two reducing equivalents, nicotinamide adenine dinucleotide (NADH) and flavin adenine nucleotide (FADH_2_), which further transfer electrons to the electronic transport chain via redox reactions [31].

Mitochondrial Ca^2+^ fulfils energy requirements by regulating mitochondrial bioenergetics [32]. Typically, an increase in mitochondrial Ca^2+^ is accompanied by increased respiration, NADH generation, and ATP production [33].

Figure 2 provides an overview of key processes in energy metabolism, including oxidative phosphorylation of the respiratory chain, glycolysis, the tricarboxylic acid cycle, and interactions with the glutathione system. Under aerobic conditions, pyruvate produced by glycolysis is introduced into the TCA cycle (via acetyl-CoA, catalyzed by pyruvate dehydrogenase with NAD^+^, or oxaloacetate, catalyzed by pyruvate carboxylase with ATP), which theoretically generates 36 ATP molecules per cycle through the respiratory chain in the mitochondria (O_2_-dependent pathway for ATP production). Under hypoxic conditions, most cells can shift their primary metabolic strategy from predominantly mitochondrial respiration toward increased glycolysis and generate ATP only through substrate-level phosphorylation (conversion of pyruvate into lactate in the cytosol and transfer of phosphate from intermediates (such as creatine phosphate)) [34,35,36].

### 4.2. ATP Use

At the whole-organism level, ATP production is made up of Na^+^/K^+^–ATPase activity (20–25%), protein synthesis (20–25%), gluconeogenesis (~7%), Ca^2+^–ATPase activity (~5%), actinomyosin–ATPase activity (~5%), ureagenesis (~2%), and all other ATP-consuming processes (~6%) [28,30].

Cellular homeostasis largely depends on ionic regulation. It should be emphasized once again, as mentioned above, that 25% of ATP production is spent only for Na^+^/K^+^–ATPase activity. It should be noted that the glutamate transporter and Na^+^/K^+^–ATPase colocalize with mitochondria and glycolytic enzymes [33]. For example, after capture by mitochondria, Ca^2+^ is released back into the cytosol through the activity of Na^+^/Ca^2+^ and H^+^/Ca^2+^ exchangers [33]. An increase in Na^+^ level promotes mitochondrial Ca^2+^ efflux, which is an important mechanism for maintaining Na^+^-Ca^2+^ balance in mitochondria [37].

Magnesium is a cofactor for enzymes that transfer phosphate groups, such as the ATPases, involved in energizing the pumps for H^+^ and Ca^2+^ and the exchanger for Na^+^ and K^+^, and most cellular Mg is associated with ATP [38]. Thus, the last step in glycolysis is the transfer of the phosphoryl group from phosphoenolpyruvate to ADP, which is catalyzed by pyruvate kinase, which requires K and either Mg or Mn [34].

The suppression of the metabolic rate may involve the controlled shutdown of processes involved in membrane ion movement. A significant decrease (approximately 65%) in the gill activity of Na^+^/K^+^–ATPase of the Amazonian cichlid (*Astronotus ocellatus*) was achieved due to post-translational modification of the Na^+^/K^+^–ATPase protein under hypoxia exposure [39]. Chronic hypoxia caused a 40% decrease in Na^+^/K^+^–ATPase activity in the brain but did not affect the liver and white muscle of goldfish (*Carassius auratus*) (champion of hypoxia tolerance) [40]. The anoxic goldfish brain downregulates Na^+^/K^+^–ATPase together with Ca^2+^ channel activity to decrease ion channel leakage and maintain membrane gradients [41].

The toxicity of metals can also suppress the metabolism rate. Monteiro et al. [42] showed that an inhibition of gill Na^+^/K^+^–ATPase activity and a loss of plasma Na^+^ and Cl^−^ were observed in *Oreochromis niloticus* during Cu exposure. Da Silva and Martineza [43] showed that Na^+^/K^+^–ATPase activity in the gills and kidneys of juvenile freshwater teleost (*Prochilodus lineatus*) was significantly decreased after 24 and 96 h of exposure to 10 μg/L Cd, whereas Ca^2+^–ATPase activity was significantly decreased only in the gills.

An electrolyte-imbalanced diet acts as a chronic stressor in rainbow trout (*Oncorhynchus mykiss*), causing a higher energy demand to maintain the acid–base balance [44].

Exposure to metals can cause additional energy costs for detoxification, control of metal homeostasis, and compensatory reactions, including ionic regulation, at both the organismal and cellular levels. For example, an intensive metabolism of Na, K, and Mg has been identified in fish as an adaptation mechanism due to high respiratory activity under both thermal pollution [45] and metal pollution [46].

### 4.3. ATP and ROS Production

Under normal physiological conditions, reactive oxygen species (ROS) emission is considered to account for ~2% of the total oxygen consumed by mitochondria (values can range from 0.25 to 11% depending on the animal species and respiration rates) [47]. The mitochondrial permeability transition pore (mPTP) opening threshold is strongly dependent on ambient molecular oxygen (since the formation of ROS in the system is a first-order reaction by molecular oxygen) and the rate of oxidant quenching (the level of antioxidants, e.g., glutathione) [47].

On the one hand, ATP production is accompanied by proton leaking and the generation of ROS; on the other hand, adenosine (a metabolite of ATP) contributes significantly to cytoprotection. Rumkumar et al. [48] provide the following arguments:-The levels of adenosine are determined primarily from the dephosphorylation of its immediate precursor, adenosine monophosphate (AMP); on the other hand, phosphorylation plays an integral role in the activation of antioxidant enzymes;-Under normal conditions, adenosine is phosphorylated by adenosine kinase to AMP and subsequently to ATP to restore the nucleotide pool, but adenosine is also produced from the hydrolysis of S-adenosylhomocysteine by S-adenosylhomocysteine hydrolase;-Adenosine receptors can be regulated by oxidative stress, and their activation leads to an increase in the activities of superoxide dismutase, catalase, glutathione peroxidase, and glutathione reductase, along with a reduction in malondialdehyde (a marker of lipid peroxidation);-Adenosine also contributes to sedation, bradycardia, vasorelaxation, the inhibition of lipolysis, and the regulation of the immune system.

On the one hand, in a review of the last 10 years of research, oxidative stress was the most recurrent effect of metal toxicity [22].

On the other hand, metabolic studies clearly show an increase in adenosine levels, such as in the case of nickel [49] and cadmium and zinc [50] exposure in the test organisms (mussels and clams).

Thus, achieving a reasonable balance between ROS and ATP production by mitochondria is crucial because it reflects the current energy requirements of the cell in a particular physiological state [47].

### 4.4. Mitochondria Are Target Organelles

The mitochondrial membrane potential (ΔΨ) is an essential attribute of mitochondria, and its homeostasis is a prerequisite for mitochondria health and the preservation of normal cell and tissue function The mitochondrial membrane potential (due to the negative charge of the mitochondrial interior) drives cations into mitochondria, and the latter accumulates these cations to a thermodynamically acceptable level (which theoretically exceeds extramitochondrial levels by about three orders of magnitude); in case of a drop in the ΔΨ below the phosphorylating membrane potential, mitochondria can initiate a process of self-destruction with the participation of Ca^2+^-independent degradative hydrolases and phospholipases [47]. Excessive mitochondrial Ca^2+^ uptake causes an increase in ROS production, ATP synthesis inhibition, mPTP opening, cytochrome c release, caspase activation, and apoptosis [33]. For example, besides cadmium inhibiting plasma membrane calcium channels and Ca^2+^–ATPase groups, gluconeogenesis, and oxidative phosphorylation, cadmium could directly lead to the dysfunction of isolated mitochondria, including the inhibition of respiration, loss of transmembrane potential, and the release of cytochrome c [51].

Indicated by immunohistochemistry, the density and distribution of chloride cells (characterized by numerous mitochondria and an extensive tubular membrane system containing a high density of Na^+^/K^+^–ATPase activity) were significantly lower in the gills of *Prochilodus lineatus* exposed to Al after 96 h, and the activity of Na^+^/K^+^–ATPase was 50% lower [52]. Hypoxia-acclimated goldfish mainly contribute to metabolic suppression by regulating the glycolytic pyruvate supply, reducing brain Na^+^/K^+^–ATPase, and downregulating cytochrome c oxidase, which likely decreases mitochondrial density [40]. After acute hypoxic stress, a study of the liver structure showed that the amount of mitochondria in hepatocytes of *Trachinotus ovatus* decreased, whereas intracellular ROS levels increased, and the antioxidant system was enhanced to attenuate oxidative damage [53]. In another study, transmission electron microscopy results also revealed that significant amounts of vacuolated mitochondria were observed in zebrafish hepatocytes under Cu exposure [54].

Under normal physiological conditions, the influx and outflow of metal ions from mitochondria are in dynamic equilibrium, but some metal ions are not necessary for mitochondria [37]. The compartmentalization of elements may play a protective role against metal toxicity. For example, the functional interaction between mitochondria and the endoplasmic reticulum (ER) deeply affects the correct mitochondrial Ca^2+^ signal, thus modulating cell bioenergetics and functionality [32]. In a study where subcellular fractions of fish liver were separated, the cytosolic heat-stable fraction (metallothionein (MT) and glutathione (GSH)) was consistently involved in the detoxification of all trace metals, and granule-like structures played a complementary role in the detoxification; however, these detoxification mechanisms were not completely effective because increasing trace metal concentrations in whole livers were accompanied by significant increases in the concentrations of most trace metals in “sensitive” subcellular fractions, that is, mitochondria, heat-denatured cytosolic proteins, and microsomes and lysosomes [55].

### 4.5. Catabolism, Hypoxia, and Anaerobic Metabolism

Catabolism is the degradative phase of metabolism; catabolic pathways release energy, some of which is conserved in the formation of ATP and reduced electron carriers (NADH and FADH_2_), and the rest is lost as heat.

REV-ERBα (a transcriptional repressor that regulates nutrient metabolism and influences energy homeostasis) plays a key role in the regulation of several physiological functions, such as energy balance and circadian rhythms [56]. To determine the involvement of REV-ERBαs in the energy balance and metabolism of fish. Saiz et al. [57] studied the effects of its agonist SR9009, which promoted a negative balance by reducing food intake and modifying the hepatic metabolism of lipids and carbohydrates (SR9009 decreased plasma glucose, significantly increased in the activity of pyruvate kinase, which coincided with increased glycolysis and decreased gluconeogenesis in the liver).

Starving rainbow trout (*Oncorhynchus mykiss*) for 4 weeks resulted in the utilization of select tissue fatty acids and increased catabolism of cellular proteins; most importantly, the gluconeogenic amino acids alanine and glutamine were significantly reduced [58]. The fish primarily mobilizes lipid stores as energy during fasting to cope with a negative energy balance; the protein and carbohydrate metabolism was also affected by fasting, as muscle protein was catabolized and mechanisms to preserve liver glycogen were initiated (maintained glycogen levels may be important as part of the metabolic adaptation during a catabolic situation) [59].

Toxic substances, including metals, may mimic the effects of environmental stressors (hypoxia, starvation) because they also cause a decrease in the rates of feeding and ventilation [60]. It is not surprising that increased blood glucose levels are a general stress response in fish [11]. Under metal stress, three freshwater fish species (common carp (*Cyprinus carpio*), silver carp (*Hypothalmic molitrix*), and tilapia (*Oreochromis niloticus*)) in the Nhue–Day River basin, Vietnam, demonstrated high energy demand (alteration in glycogen reserves, protein catabolism, changes in the activity of antioxidant enzymes to provide tolerance to oxidative stressors) [61]. For example, Cu exposure can disturb the normal processes of lipid metabolism and affect the initial stages of the TCA cycle; the lipid content, activities of lipogenic enzymes, and isocitrate dehydrogenase activity were reduced in the liver of juvenile yellow catfish *Pelteobagrus fulvidraco* [62].

Under hypoxic conditions (when oxygen consumption is below basal level), some physiological functions may be suppressed and oxygen debt will accumulate; moreover, replacing aerobic metabolism with anaerobic glycolysis is only a temporary solution because of a less efficient use of fuel and an accumulation of toxic wastes [29]. Under hypoxic conditions, when mitochondria are incapable of sustaining ΔΨ driven by respiration, the mitochondrial membrane potential is maintained at the expense of cytosolic ATP hydrolysis [47]. Adenosine levels increase significantly following metabolic insults such as hypoxia [48], which also leads to a decrease in ATP levels.

Adaptation to hypoxia can occur in various ways, including the reversible remodeling of the gill structure to increase the area of contact with the aquatic environment for oxygen exchange [63,64,65,66] and an increase in red blood cells and hemoglobin in erythrocytes [67,68]. For example, the strategy of whitefish (*Coregonus lavaretus* L.) from the historically contaminated area was aimed at proliferation of the gill structure, which increased their functional surface and reduced the distance to the bloodstream, as well as increasing hemoglobin in maturing erythrocytes [46]. However, adaptation to hypoxia is also associated with a restructuring of the metabolism to meet energy demands and maintain intracellular homeostasis [53,68].

Under hypoxia, the metabolic switch from predominantly mitochondrial respiration toward increased glycolysis can be regulated by several pathways, including the hypoxia inducible factor-1α (HIF-1α), which induces an increased expression of glycolytic enzymes [36]. Hypoxia can also induce the AMPK/HIF-1α/HSP70 pathway; as AMP-activated protein kinase (AMPK) phosphorylates and inhibits acetyl-CoA carboxylase, liver fatty acid biosynthesis is uniformly inhibited, and the observed activation of AMPK may be required to initiate the protective upregulation of HSP70 (heat shock proteins are vital to maintain protein functionality under stress and cellular dysfunction, as the accumulation of misfolded proteins is energetically expensive and cytotoxic) and the suite of physiological responses [68].

Hypoxia survival in the hypoxia-tolerant killifish (*Fundulus heteroclitus*) involves the transient activation of substrate-level phosphorylation (decreases in creatine phospate and increases in lactate); in addition, estimated changes in cytoplasmic and mitochondrial NAD^+^/NADH did not parallel one another and the mitochondrial NADH shuttles did not function during hypoxia exposure, and the inactivation of pyruvate dehydrogenase limited the oxidation of mitochondrial pyruvate [35]. A metabolic study revealed an upregulation of lactate, alanine, and creatine phosphate in moderately hypoxia-tolerant fish, the common carp (*Cyprinus carpio*) [69]. Proteomics and metabolomics analysis of the seabream (*Sparus aurata*) liver revealed different pathways in metabolic regulation: pyruvate carboxylase was downregulated in hypoxia-exposed fish, together with the upregulation of lactate and creatine phosphate, and the downregulation of isocitrate dehydrogenase also corroborates the suppression of this pathway, viz., the TCA cycle, together with the upregulation of glucogenic amino acids (e.g., alanine, arginine, proline), which can further produce glucose [70].

In studies of the effects of hypoxia, the accumulation of succinate (an intermediate of the TCA cycle, the end product of anaerobic metabolism) was observed, which is a clear sign of facultative anaerobiosis [60,71]. Significant increases in succinate levels were observed with nickel [49], cadmium and zinc [50], and cadmium and copper [72] exposure in the test organisms (mussels and clams).

The switch from aerobic to anaerobic metabolism is considered a physiological adaptation when exposed to various stressors, and the diagnosis of this switch can be based on biomarkers of anaerobic and aerobic energy production pathways (as important indicators of energetic demands); for example, if lactate dehydrogenase (LDH) plays a key role in anaerobic pathways (responsible for the conversion of lactate back to pyruvate in the liver), then isocitrate dehydrogenase (IDH), involved in energy production through aerobic pathways, and which catalyzes a key step in the TCA cycle, is also involved in antioxidant defense, where it is critical for the regeneration of NADH, required for glutathione conjugation pathways [73]. It should also be noted that creatine kinase is a controller of energy metabolism in tissues with a large energy demand, and it is highly susceptible to inactivation by free radicals and oxidative damage [74].

Cho et al. [75] examined the crystal structure of NAD^+^-dependent cytosolic IDH in the presence of Cd^2+^ and showed that Cd^2+^ inactivates IDH due to its high affinity for thiols, leading to the covalent modification of the enzyme; however, residual Cd^2+^ also activates IDH, providing the divalent cation required for catalytic activity. Although reducing agents (e.g., glutathione) cannot restore activity after Cd^2+^ inactivation, they can maintain IDH activity by chelating Cd^2+^. Thus, on the one hand, in the presence of Cd^2+^-consuming cellular antioxidants, cells must constantly supply reducing agents to protect against oxidative stress; on the other hand, the ability of IDH to use Cd^2+^ to generate NADH may allow cells to protect themselves from Cd^2+^. It should also be noted that the inhibition of IDH activity is more important for antioxidant defense than the inhibition of succinate dehydrogenase (SDH) activity.

Perumalsamy et al. [76] demonstrated that the enzyme profile of *Oreochromis mossambicus* exposed to a mixture of metals (electroplating industry wastewater) was greatly altered, with a significant increase in LDH levels and a sharp decrease in SDH levels in fish liver. Exposure to titanium dioxide nanoparticles altered the metabolic function of the liver and impaired flatfish bioenergetics, suppressing aerobic hepatic metabolism by increasing LDH activity while decreasing SDH activity, and most importantly, depleting IDH activity (NAD^+^-dependent) [77].

Thus, physiological defense mechanisms against metal toxicity cause increased energy use, which seems to impair oxidative metabolism and increase anaerobic metabolism.

## 5. Conclusions and Future Perspectives

Metal toxicity in the aquatic environment is an ongoing problem due to both dispersion from natural sources and inevitable input from anthropogenic activities. Moreover, the chronic effects of polymetallic pollution on aquatic organisms are becoming increasingly widespread, and it is more difficult to interpret the effects.

Redox reactions are “the foundation-stone” of ATP production, ROS emission, the activation of antioxidant enzymes, and the effects of metal toxicity. The considered mechanisms of regulation of ATP production and ROS emission show processes for achieving their reasonable balance while maintaining normal functions of cells and tissues. Mitochondria can also be target organelles for metal toxicity, especially when the mitochondrial membrane potential drops below its phosphorylation level.

Beyond physiological needs, additional energy costs can be considered as an additional measure of the impact of metal toxicity on survival. Moreover, metals can imitate hypoxic conditions, disrupting oxidative metabolism and enhancing the anaerobic metabolism.

The metabolic approach can provide a unifying framework (as the boundary conditions of survival) to connect many laboratory experiments and field studies on the effects of metal toxicity on aquatic organisms.

## Figures and Tables

**Figure 1 ijms-25-05015-f001:**
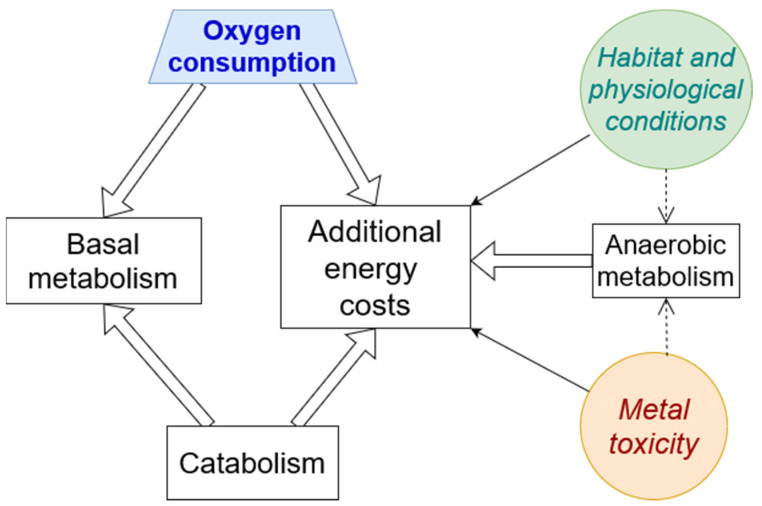
General diagram of energy flows and the conditions affecting them (circles). Thick solid arrows show oxygen inflows and energy flows, thin solid arrows show effects on these flows, and thin dashed arrows indicate the possibility of the effects.

**Figure 2 ijms-25-05015-f002:**
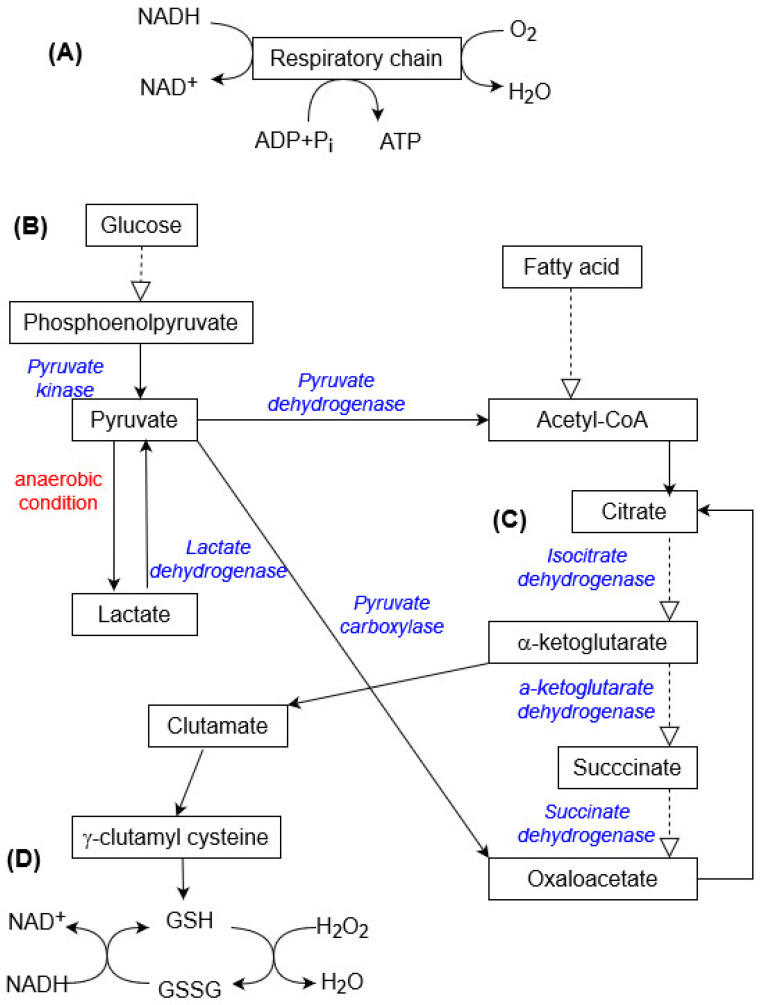
Overview of the most important metabolic pathways: respiratory chain (**A**), glycolysis (**B**), tricarboxylic acid cycle (**C**), and glutathione system (**D**) (GSH—glutathione, GSSG—oxidized glutathione). The dashed arrows show multistep reactions.

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
