# Peer review of "Metal Toxicity: Effects on Energy Metabolism in Fish"

_ijms, 2024, doi:10.3390/ijms25095015_

Round 1

Reviewer 1 Report

Comments and Suggestions for Authors

See attached PDF with comments. 

Comments on the Quality of English Language

The English is not of great quality. It is good enough to be generally understood but needs proper revision prior to publishing. 

Author Response

Response to Reviewer 1 Comments

The author is deeply appreciated for the careful consideration of the manuscript and comments. Detailed responses to each comment are given below. Corrections in the manuscript are highlighted in red.

Point 1: This review is in my opinion of low interest. The topic should be about mechanisms of toxicity of metals on fish energy metabolism. However, much of the text discusses other factors (hypoxia, temperature) and then only mention similarities with metal toxicity or additional interactions of metals. Some examples are not about fish (bivalves for instance).

Response 1: The review emphasizes the difficulty of assessing the effects of metal mixtures under natural conditions. Therefore, effects on energy metabolism are considered from different viewpoints. Temperature and mussels are given very little attention, but these examples show the commonality of responses. Hypoxia is given more attention because metals disrupt the processes of normal aerobic matabolism.

Point 2: Line 31.This statement is unclear to me.

Response 2: The clarification has been included in the text (lines 37-38).

Point 3: Line 36. Incomplete? Rewrite.

Response 3: The clarification has been included in the text (line 46).

Point 4: Line 87. do  you mean duration?

Response 4: The corrections have been done (line 99).

Point 5: Lines 94, 96. disturbances?

Response 5: The corrections have been done (lines 106, 108).

Point 6: Line 98.This is too general.

Response 6: This aspect has been removed.

Point 7: Line 115. Figure #. Oxygen. What is the meaning of the various arrow types? Фигура #. Кислород. Что означают различные типы стрелок?

Response 7: A description of the different types of arrows has been added to the figure caption (lines 128-130).

Point 8: Line 122. Other substrates as well.

Response 8: Yes, you are absolutely right, other substrates are also used in energy production, but amino acids and lipids enter at various stages in the TCA cycle, and glucose undergoes glycolysis.

Point 9: Line 145. Unclear. Rewrite.

Response 9: Rewritten (lines 161).

Point 10: Lines 214-215. I don't see how the second part of this statement explains why it's crucial.

Response 10: The quenching of oxidants is also an energy-consuming process, so emphasis is placed on the body's capabilities.

Point 11: Lines 235, 240. Spell out in full at first use and indicate what it is (is it a fish for instance?). Apply to all species listed in the text.

Response 11: The names of species are now given in full (lines 252, 257).

Point 12: Lines 238-243. These statements seem unrelated, mentioning different responses to different stressors to different organisms.

Response 12: Yes, you are right, they are different studies. A clarification has been made in the text (lines 258-259).

Point 13: Lines 258-260. This is textbook knowledge. Not sure it needs a reference.

Response 13: The reference is omitted as you suggested (line 279).

Point 14: Lines 353-255. No such conclusion can be drawn from Figure 2.

Response 14: The link to the figure has been removed (line 375).

Point 15: Lines 364-365. It is the case in some studies, but several other studies did not report this.

Response 15: The phrase is betrayed by its probabilistic character (line 384).

Reviewer 2 Report

Comments and Suggestions for Authors

Heavy metals are environmental pollutants with non-degradability, toxicity, and bioaccumulation to aquatic organisms (including fish). Accumulated heavy metals will disturb normal physiological/metabolic processes and even cause mortality. Metal toxicity exactly impacts energetic metabolism in fish and is valued to be reviewed for highlighting the corresponding bad consequences as this manuscript (MS) did. Nevertheless, major revisions are necessary for the manuscript throughout. Following are my comments and suggestions:

1.     The authoritative and professional definitions are absolutely necessary for several important terms in suitable locations of the MS, especially, metal, heavy metal, macro-/micro- elements, essential metal, toxic element, as well as energetic metabolism/energy metabolism. Actually, chaos and confusion are caused in the MS by unsuitablely mixed use without clear definitions.

2.     Please refer clearly to which kind metal toxicities were studied in each literature cited rather than mention the ambiguous word “metal”. In now status of the MS, very few metals (only Cu, Cd and especially Ca-too many contents) were exactly mentioned the corresponding functions. However, those of Ca were essential and were not toxic consequences. Therefore, the author need to re-provide effective information of more certain species of heavy metals in the MS according to the title of the MS.

3.     Many literature cited might be the general information of oxygen consumption, ATP production, catabolism, hypoxia, anaerobic metabolism that might be without direct relation to metal-caused or caused by other type pollutants. I suggest it is better for the authors to delete these references and re-cite those related directly to the theme of the MS.  

Author Response

Response to Reviewer 2 Comments

The author is deeply appreciated for the careful consideration of the manuscript and comments. Detailed responses to each comment are given below. Corrections in the manuscript are highlighted in red.

Point 1: The authoritative and professional definitions are absolutely necessary for several important terms in suitable locations of the MS, especially, metal, heavy metal, macro-/micro- elements, essential metal, toxic element, as well as energetic metabolism/energy metabolism. Actually, chaos and confusion are caused in the MS by unsuitablely mixed use without clear definitions.

Response 1: Indeed, definitions of some important terms are necessary and have been included in the text (lines 23-27, 33-34, 40-43).

Point 2: Please refer clearly to which kind metal toxicities were studied in each literature cited rather than mention the ambiguous word “metal”. In now status of the MS, very few metals (only Cu, Cd and especially Ca-too many contents) were exactly mentioned the corresponding functions. However, those of Ca were essential and were not toxic consequences. Therefore, the author need to re-provide effective information of more certain species of heavy metals in the MS according to the title of the MS.

Response 2: In addition to general functions, specific functions for a number of metals have been added (lines 110-112), but the review emphasizes the difficulty of assessing exposure to mixtures of metals under natural conditions, so energy metabolism is considered as a measure of the toxic load on the organism. Calcium is given much attention, not for toxicity, but for its exclusive role in regulating energy metabolism, whereas toxic metals can impair mitochondrial transport, which is the focus of the section "4.4 Mitochondria Are Target Organelles".

Point 3: Many literature cited might be the general information of oxygen consumption, ATP production, catabolism, hypoxia, anaerobic metabolism that might be without direct relation to metal-caused or caused by other type pollutants. I suggest it is better for the authors to delete these references and re-cite those related directly to the theme of the MS.

Response 3: The effects of metals and their mixtures on energy metabolism have been considered from various viewpoints, so information on oxygen consumption, ATP production, catabolism, hypoxia, and anaerobic metabolism is essential.

Round 2

Reviewer 2 Report

Comments and Suggestions for Authors

The authors have almost addressed my concerns in the revised manuscript.  Still, I would like to provide several most recent literatures for the author to improve the revised MS, e.g., (1) Lall, S.P.; Kaushik, S.J. Nutrition and Metabolism of Minerals in Fish. Animals 2021, 11, 2711. (2) Naz, S.; Chatha, A.M.M.; Téllez-Isaías, G.; Ullah, S.; Ullah, Q.; Khan, M.Z.; Shah, M.K.; Abbas, G.; Kiran, A.; Mushtaq, R.; et al. A Comprehensive Review on Metallic Trace Elements Toxicity in Fishes and Potential Remedial Measures. Water 2023, 15, 3017. (3) Jamil Emon, F.; Rohani, M.F.; Sumaiya, N.; Tuj Jannat, M.F.; Akter, Y.; Shahjahan, M.; Abdul Kari, Z.; Tahiluddin, A.B.; Goh, K.W. Bioaccumulation and Bioremediation of Heavy Metals in Fishes—A Review. Toxics 2023, 11, 510.

Author Response

Response to Reviewer 2 Comment

The author is deeply appreciated for the comment. Corrections in the manuscript are highlighted in red.

Point 1: The authors have almost addressed my concerns in the revised manuscript.  Still, I would like to provide several most recent literatures for the author to improve the revised MS, e.g., (1) Lall, S.P.; Kaushik, S.J. Nutrition and Metabolism of Minerals in Fish. Animals 2021, 11, 2711. (2) Naz, S.; Chatha, A.M.M.; Téllez-Isaías, G.; Ullah, S.; Ullah, Q.; Khan, M.Z.; Shah, M.K.; Abbas, G.; Kiran, A.; Mushtaq, R.; et al. A Comprehensive Review on Metallic Trace Elements Toxicity in Fishes and Potential Remedial Measures. Water 2023, 15, 3017. (3) Jamil Emon, F.; Rohani, M.F.; Sumaiya, N.; Tuj Jannat, M.F.; Akter, Y.; Shahjahan, M.; Abdul Kari, Z.; Tahiluddin, A.B.; Goh, K.W. Bioaccumulation and Bioremediation of Heavy Metals in Fishes—A Review. Toxics 2023, 11, 510.

Response 1: I am grateful to the reviewer for the proposed articles, which supplemented the manuscript. The corrections have been done (lines 25-28, 103, 113-114).
